# Utility of Superb Microvascular Imaging in the Assessment of Foot Perfusion in Patients with Critical Limb Ischemia

**DOI:** 10.3390/diagnostics12112577

**Published:** 2022-10-24

**Authors:** Yuta Suto, Wakana Sato, Takayuki Yamanaka, Mayu Unuma, Yuki Kobayashi, Mako Aokawa, Hiroyuki Watanabe

**Affiliations:** Department of Cardiovascular Medicine, Akita University Graduate School of Medicine, Akita 010-8543, Japan

**Keywords:** foot perfusion, superb microvascular imaging, critical limb ischemia

## Abstract

(1) Background: Although the ankle–brachial index (ABI) and skin perfusion pressure (SPP) are commonly used to evaluate the peripheral circulation in critical limb ischemia (CLI), they often cannot be performed on sore areas. We investigated the utility of superb microvascular imaging (SMI) for assessing foot perfusion in CLI patients. (2) Methods: We measured the SMI-based vascular index (SMI-VI) at six sites in the foot before and after endovascular treatment (EVT) in 50 patients with CLI who underwent EVT of the superficial femoral artery and compared the results with SPP values and the ABI. (3) Results: SMI visualized foot perfusion in all subjects in accordance with the angiosome, including the toe areas, while the ABI was unmeasurable in three patients on hemodialysis and SPP failed in four patients. SMI-VI values were significantly lower in the CLI group than in controls, and the plantar SMI-VI had the highest diagnostic performance for CLI (sensitivity 88.6%, specificity 95.6%). After EVT, the increase in the SMI-VI was positively correlated with the increase in SPP but not that in the ABI, implying that the SMI-VI reflects foot microcirculation. (4) Conclusions: SMI enables the visualization and quantification of foot microcirculation based on the angiosome. SMI has high utility as a tool for assessing foot perfusion in CLI.

## 1. Introduction

Approximately 202 million people worldwide are affected by peripheral arterial disease (PAD) [1], which usually develops after the age of 50 years and increases exponentially after the age of 65 years. The total number of individuals with PAD is booming, with a 23% increase in the last decade as a result of the total population increase, global aging, the increased incidence of diabetes worldwide, and smoking in low- and middle-income countries [2]. Clinical symptoms of PAD progress from intermittent claudication, rest pain, and skin ulcers to gangrene of the limbs. Critical limb ischemia (CLI), which is defined by the presence of rest pain, skin ulcers, or gangrene, is the most severe stage of PAD and is associated with high rates of mortality and amputation [3]. The prevalence of CLI is approximately 0.4%, with an estimated annual incidence of 500–1000 new cases per million [4]. Because CLI is associated with high short-term risks of limb loss, cardiovascular events, and stroke, there is a need for a method to assess the affected microcirculation and thus determine a prompt course of treatment [5]. Recently, the assessment of foot microcirculation based on six angiosomes was recommended for the management of patients with CLI [6,7]. According to the angiosome concept, the foot can be divided into six angiosomes: three arise from the posterior tibial artery, one from the anterior tibial artery, and two from the peroneal artery. The anterior tibial artery supplies the dorsal side of the toe and dorsal foot; the posterior tibial artery supplies the plantar side of the toe, the web space of the toes, the plantar foot, and the inside of the heel; and the peroneal artery supplies the lateral ankle and outer aspect of the heel [8]. This concept of six angiosomes provides practical information concerning vascular anatomy; thus, it facilitates reconstruction and vascular surgery in the treatment of CLI [9,10].

Lower-limb ischemia is often diagnosed based on an ankle–brachial index (ABI) value < 0.90. The ABI is based on the systolic blood pressure at the ankle divided by the systolic blood pressure at the arm. However, its utility in patients with less severe stenosis or calcified vessels is unclear, and it does not provide information concerning the state of blood flow below the ankle. Skin perfusion pressure (SPP) is a frequently used marker of peripheral circulation in the foot, but its measurement is often hindered by pain or movement artifacts [11]; moreover, it can be difficult to measure at most ischemic sites. Thus, for a detailed evaluation of circulation in the lower limbs, better imaging modalities are needed.

Superb microvascular imaging (SMI; Canon Medical Systems, Otawara, Japan) is a new ultrasound technique that expands the visible range of blood flow and enables the visualization of extremely low-velocity flow that cannot be detected by ultrasound. Compared with conventional Doppler technologies, the advantages of SMI include its high frame rates, high resolution, high sensitivity, and reduced number of motion artifacts. These advantages allow the evaluation of tissue perfusion, especially in areas near the surface of the body, therefore, SMI may be the optimal modality for the evaluation of skin and subcutaneous tissue perfusion. In practice, SMI has been used to evaluate thyroid nodules, breast tumors, and focal liver lesions [12,13,14]; however, its utility in peripheral arterial disease has not been determined. In this study, we investigated the use of SMI to assess foot perfusion in patients with CLI, and we explored the practical value of this technique as a diagnostic aid.

## 2. Materials and Methods

### 2.1. Study Design

This prospective observational study was conducted at Akita University Hospital from 1 April 2018 to 31 March 2020. Akita University Hospital is a tertiary care hospital that serves a population of approximately 1 million people and provides treatment for complex cardiovascular diseases. The study protocol was approved by the Institutional Review Board of the Akita University Graduate School of Medicine (no. 2089). Informed consent was obtained from all study participants.

### 2.2. Participants

The 54 consecutive patients with CLI enrolled in the study had at least 90% stenosis or occlusion of the superficial femoral artery (SFA). Additionally, 42 of them also had below-the-knee arterial stenoses. A diagnosis of CLI was defined as an ABI < 0.90 or SPP < 50 mmHg, together with pain at rest and ulceration or necrosis of a lower limb for >2 weeks [15]. All patients in the CLI group had lower-limb arterial stenosis or occlusion, as demonstrated by Doppler ultrasonography, computed tomography, or transcatheter angiography; they had agreed to undergo endovascular treatment (EVT) of the SFA. Patients with uncontrolled sepsis and unsuccessful EVT were excluded. Additionally, 35 healthy volunteers with no history of cardiovascular disease were recruited as a control group.

### 2.3. Data Collection

The following demographic and clinical data were recorded: age, sex, smoking history, clinical symptoms, comorbidities (hypertension, dyslipidemia, diabetes mellitus, chronic kidney disease, and hemodialysis), and medications. Smoking history was considered to include past or current smoking. Symptoms were classified according to ischemic findings as pain at rest, ulceration, or necrosis. In patients with ulceration or necrosis, the location was recorded. Hypertension was recorded when patients used medications for hypertension, or when they had a systolic blood pressure > 140 mmHg and/or a diastolic blood pressure > 90 mmHg [16]. Dyslipidemia was recorded when patients used medications for dyslipidemia, or when they had a low-density lipoprotein cholesterol concentration > 140 mg/dL and/or a high-density lipoprotein cholesterol concentration < 40 mg/dL and/or a triglyceride concentration > 150 mg/dL [17]. Diabetes mellitus was recorded when patients used medications for diabetes mellitus, or when they had a fasting plasma glucose level > 126 mg/dL and/or casual plasma glucose level > 200 mg/dL and/or HbA1c > 6.5% [18]. Chronic kidney disease was recorded when patients had an estimated glomerular filtration rate < 60 mL/min/1.73 m^2^ [19].

### 2.4. Superb Microvascular Imaging and Vascular Index

Conventional Doppler imaging cannot be used in the clinical assessment of the microvascular system because it cannot distinguish low-flow blood signals from tissue movement artifacts (clutter) [20]. By contrast, SMI uses an adaptive wall filter to suppress clutter and preserves low-flow signals [21]. Consequently, SMI can detect very low-velocity blood flow and evaluate microcirculation.

Microcirculation was quantitively assessed by measuring an SMI-based vascular index (SMI-VI, %). The SMI-VI is defined as the ratio of Doppler signal pixels to total pixels in the region of interest; thus, it describes the microvascular areal bed proportion (Figure 1A). In the absence of reports concerning the use of SMI to assess foot microcirculation, this study used the following SMI-VI measurement protocol. Each participant was placed at rest in a supine position; the evaluator set the depth of observation to 1.5 cm below the skin surface, and the region of interest was regarded as 0.5 cm × 1.5 cm. The SMI-VI was measured during the maximum-blood-flow phase of the cardiac cycle. SMI-VI measurement points were localized in accordance with the concept of six angiosomes: dorsal, plantar, median heel, lateral heel, thumb, and little finger (Figure 1B). The plantar SMI-VI was calculated as the mean of three sites: medial side, lateral side, and heel. To provide reproducibility when measuring the same patient multiple times, the probe positioning at each measurement point was marked. In the CLI group, SMI-VI was measured on the affected limb. In patients and controls, SMI-VI was acquired using an Aplio i900 (Canon Medical Systems Corp.). An iDMS PLI-2004 BX probe (24 MHz) was used in the foot area, and a PLI-2002 BT probe (22 MHz) was used in the toe area. Evaluators were limited to two specific individuals: one doctor and one clinical-laboratory technician. The temperature of the laboratory was maintained at 25 °C.

### 2.5. Measurement of ABI and SPP

In addition to the SMI-VI, the ABI and SPP were measured in the CLI group.

The ABI was measured using a VS-3000 E (Fukuda Denshi, Tokyo, Japan). With the patient in a supine position, brachial and ankle blood pressures were measured in both left and right limbs. The ABI was then calculated through the division of the ankle pressure determined on each leg by the highest brachial systolic pressure [22]. The ABI on the affected side was used in this study. Discontinuation criteria for ABI measurement were severe pain during blood pressure measurement and a flat ankle pulse wave.

The SPP was measured using a SensiLase PAD 3000 (Vasamed, Eden Prairie, MN, USA). With the patient in a supine position, measurements were obtained at the dorsal and plantar sites of the affected limb. A laser sensor and a blood pressure band (cuff) were wrapped around the measurement site. The cuff was inflated to interrupt blood flow, then gradually deflated to determine the pressure at which skin microcirculation resumed [23]. Discontinuation criteria for SPP measurement were severe pain caused by cuff expansion and unremovable artifacts resulting from involuntary movement of the foot.

### 2.6. Statistical Analysis

Continuous variables are presented as medians and interquartile ranges (IQRs) or as means and standard deviations; categorical variables are presented as numerals and percentages. The SMI-VI values of the CLI and control groups were compared using Student’s *t*-test. Receiver operating characteristic (ROC) curve analysis was conducted to test the diagnostic performance of the SMI-VI. SMI-VI, ABI, and SPP values before and after EVT in the CLI group were compared using paired-samples *t*-tests. Scatter plots were created to compare changes in the ABI, SPP, and SMI-VI after EVT. The correlations between these values were analyzed using the Pearson product-moment correlation coefficient. Positive correlations were classified as poor, r = 0–0.2; weak, r = 0.2–0.4; moderate, r = 0.4–0.7; or strong, r = 0.7–1.0. A subanalysis of the SMI-VI and the ABI in the hemodialysis and non-hemodialysis groups was performed using the same method. A *p*-value < 0.05 was considered statistically significant. Missing data were not compensated in the analysis. All analyses were performed using IBM SPSS version 28.0 (IBM Corp., Armonk, NY, USA) and GraphPad Prism version 7.03 (GraphPad Software, San Diego, CA, USA).

## 3. Results

### 3.1. Patient Characteristics

Among the 54 patients with CLI, one was excluded because of uncontrolled sepsis and three were excluded because of unsuccessful EVT. The remaining 50 patients (50 limbs) were included in the CLI group. Table 1 summarizes the patients’ characteristics. The 36 men and 14 women had a median age of 73 years (IQR: 67−79 years). All patients had leg pain at rest. Thirteen patients had skin and soft-tissue wounds: five had ulcers and eight had necrosis. Of the patients with soft-tissue wounds, four lesions were on the toes, two on the heel, one on the ankle, one on the shin, and five were in multiple locations. The comorbidities were diabetes mellitus in 30 patients and hemodialysis (HD) in 11 patients.

The 19 men and 16 women in the control group had a median age of 49 years (IQR: 38–65 years); five had a history of smoking, six had hypertension, three had dyslipidemia, and three had diabetes mellitus. The absence of atherosclerosis in the lower extremities in the control group was confirmed by the ABI and Doppler ultrasonography.

### 3.2. Comparison of Foot SMI-VI Values between the CLI and Control Groups

Comparison of foot SMI-VI values between the CLI and control groups (Figure 2A) showed that they were significantly lower in the CLI group at each measurement point: dorsal, 4.0 ± 0.4% vs. 8.2 ± 0.6% (*p* < 0.01) (Figure 2(Aa)); plantar, 1.8 ± 1.0% vs. 5.3 ± 1.7% (*p* < 0.01) (Figure 2(Ab)); median heel, 4.0 ± 0.8% vs. 7.8 ± 0.6% (*p* < 0.01) (Figure 2(Ac)); lateral heel, 3.4 ± 0.3% vs. 6.9 ± 0.6% (*p* < 0.01) (Figure 2(Ad)); thumb, 4.2 ± 0.4% vs. 8.3 ± 0.3% (*p* < 0.01) (Figure 2(Ae)); and little finger, 4.9 ± 0.6% vs. 8.4 ± 0.2% (*p* < 0.01) (Figure 2(Af)). These results were as expected because multiple stenoses or occlusions including the SFA lesion were present in all of the patients in the CLI group and none of the participants in the control group.

### 3.3. Comparison of ABI, SPP and SMI-VI Values in CLI

Among the 50 patients with CLI, the ABI could not be measured in six because of pain (three patients) and flat pulse waves (three patients). The SPP could not be measured in nine patients because of pain (four patients) and motion artifacts (five patients). Therefore, the ABI was measured in 44 patients (88%) and the SPP was measured in 41 patients (80.5%). In contrast, the SMI-VI was measured rapidly and painlessly in all 50 patients (100%).

ROC curves were generated using the SMI-VI data from the CLI and control groups to assess the diagnostic performance of the SMI-VI. Among the measurement points, diagnostic performance was highest at the plantar site. Based on the results of the ROC curve analysis, the cut-off value of the plantar SMI-VI was 3.4%, with an area under the ROC curve of 0.969, a sensitivity of 88.6%, and a specificity of 95.6% (Figure 2B). For patients in the CLI group diagnosed based on an ABI < 0.90 or SPP < 50 mmHg, the ABI had a sensitivity of 78.0% and the SPP had a sensitivity of 82.9%. At an SMI-VI cut-off value of 3.4%, the sensitivity of the SMI-VI was comparable with those of ABI and SPP.

Because the ABI tends to be less accurate, leading to false-negative results in HD patients, we conducted a subgroup analysis of HD (*n* = 11) and non-HD (*n* = 39) CLI patients and then compared the diagnostic performance between the plantar SMI-VI and the ABI before EVT in the two groups. In the HD group, the ABI could not be determined because of pain in two patients and a flat wave in one patient. Of the eight patients whose ABI could be determined, four had ABI > 0.90, implying false negatives. However, plantar SMI-VI values were at the lowest 3.4% in all 11 HD patients. These results show that within the HD group, a cut-off value indicative of limb ischemia was acquired in only 4 of 11 patients using the ABI (36.4%) but in all patients (100%) using the SMI-VI. In the non-HD patients, the ABI could not be determined because of pain in one patient and a flat wave in two patients. One patient had an ABI value of 1.01, implying a false negative. Plantar SMI-VI values were determined for all non-dialysis patients; 2 of 39 patients had a plantar SMI-VI ≥ 3.4%, while the remaining 37 had SMI-VI < 3.4%. From these results, a cut-off value could be acquired in 35 of 39 patients using the ABI (89.7%) and in 37 of 39 patients (94.9%) using the SMI-VI in the non-HD group. In short, the SMI-VI was able to assess limb ischemia in HD patients in whom ABI assessment was difficult.

### 3.4. Changes in Foot Perfusion after EVT Evaluated by Each Modality

Patients in the CLI group underwent EVT of the SFA, with the SMI-VI measured before and 2 days after treatment. Figure 3A shows the post-treatment changes in the SMI-VI. Consistent with the known efficacy of EVT of the SFA, SMI-VI values increased significantly at all measurement points after treatment, including the dorsal area (from 4.1 ± 3.0% to 6.3 ± 3.6%, *p* = 0.0012; Figure 3(Aa)), plantar area (from 1.8 ± 1.0% to 4.0 ± 1.8%, *p* < 0.01; Figure 3(Ab)), median heel (from 2.8 ± 2.0% to 6.2 ± 3.9%, *p* < 0.01; Figure 3(Ac)), lateral heel (from 3.5 ± 2.3% to 6.6 ± 4.6%, *p* < 0.01; Figure 3(Ad)), thumb (from 4.1 ± 3.0% to 6.8 ± 4.1%, *p* < 0.01; Figure 3(Ae)), and little finger (from 5.1 ± 4.1% to 6.9 ± 4.1%, *p* < 0.01; Figure 3(Af)).

Both the ABI and SPP were measured before and after EVT; the results were compared with those based on the SMI-VI. For SPP and the SMI-VI, plantar measurement data were used to ensure consistency with the ABI. Figure 3B shows the changes in ABI, plantar SPP, and plantar SMI-VI values before and after EVT. The ABI increased from 0.62 ± 0.15 to 0.85 ± 0.25 (*p* < 0.01), plantar SPP increased from 40.6 ± 21.9 mmHg to 60.3 ± 25.1 mmHg (*p* < 0.01), and plantar SMI-VI increased from 1.8 ± 1.0% to 4.0 ± 1.8% (*p* < 0.01). The significant increases in the SMI-VI followed a pattern similar to the increases in the ABI and SPP.

The ABI is a marker of macrocirculation, whereas the SPP is a marker of microcirculation. Because the SMI-VI reflects blood flow in microvessels, we hypothesized that the SMI-VI would correlate more strongly with the SPP than with the ABI. Therefore, we calculated the differential values of the ABI, plantar SPP, and plantar SMI-VI before and after EVT: ΔABI, ΔSPP, and ΔSMI-VI. Figure 4 shows the correlations of ΔSMI-VI with ΔABI and ΔSPP. ΔSMI-VI was not significantly correlated with ΔABI (*p* = 0.27, r = 0.16), but it was positively correlated with ΔSPP (*p* = 0.02, r = 0.40).

### 3.5. Improvement in Microcirculation after EVT and Wound Healing

Wound-tissue microcirculation was measured using SMI in patients with ulcerations or necrosis who underwent EVT. The results obtained in a patient with a black necrotic ulcer on the heel are shown in Figure 5. Although there were almost no microvessels before EVT, improved microcirculation at the wound bed was detected immediately after EVT (Figure 5A). Twenty days after treatment, the necrotic tissue was debrided and the ulcer exhibited gradual healing. Abundant microcirculation was maintained within the ulcer (Figure 5B). Wound healing was not achieved in patients whose wound-tissue SMI-VI values showed little improvement after EVT. These results imply that SMI can be used to predict wound healing.

## 4. Discussion

To summarize the main findings of our study: SMI-VI values at all measurement points in the foot, including the toe area, were significantly lower in the CLI group than in the control group, implying that the SMI-VI can detect insufficient foot perfusion in patients with CLI; the SMI-VI can be used to assess foot perfusion accurately in HD patients in whom ABI assessment is difficult; the use of the SMI-VI is possible even in patients in whom SPP cannot be measured due to pain or involuntary movements; the significant increase in the SMI-VI after EVT mirrored similar increases in the ABI and SPP; ΔSMI-VI was strongly correlated with ΔSPP, but not ΔABI, implying that the SMI-VI can be used to evaluate microcirculation in the foot; and SMI values may predict wound healing.

Our study demonstrated the diagnostic value of SMI for the assessment of limb ischemia. To date, lower-limb circulation has been evaluated only in a qualitative manner, using conventional modalities such as the ABI and SPP [24,25]. By allowing the direct observation of microvessels, SMI can be used for quantitative evaluations. Data from the ROC curve analysis imply that the SMI-VI has high sensitivity and specificity for detecting limb ischemia, and its diagnostic performance compares favorably with the ABI and SPP (Figure 2B). Furthermore, as shown in Figure 4, the strong correlation of the SMI-VI with SPP rather than with ABI values implies that the SMI-VI reflects the microcirculation more strongly than the macrocirculation.

The advantage of SMI is that very localized sites, such as the toe ends, can be examined. In patients with CLI, pain at rest typically involves the toes, with ulcers usually first appearing in peripheral areas such as the toes and heel [26,27]. The assessment of perfusion in the toes is important for the management of CLI, but this is difficult with the ABI and SPP. The results of our study imply that SMI is the optimal modality for evaluating perfusion, particularly in the toe region (Figure 2A and Figure 3A).

The more accurate assessment of foot perfusion using SMI than with the ABI was particularly striking in HD patients [28,29]. In patients with reduced vascular compliance because of advanced arterial calcification, a cuff pressure greater than blood pressure may not block blood flow, resulting in a false negative based on the ABI [25,30]. In this study, ABI-based assessment was difficult in 7 of the 11 HD patients because of pain, a flat wave, or a false negative, whereas the plantar SMI-VI values of all 11 were low, implying limb ischemia. Presumably, the reduced accuracy of the ABI in HD patients is due to severe calcification of the lower-limb arteries. These results imply that SMI can be used to assess foot perfusion, even in patients with severely calcified arterial lesions, for whom the ABI is often unreliable.

SMI also enables the assessment of foot microcirculation in a manner that is less invasive than when using SPP. Other disadvantages of using SPP values include a long measurement time, pain related to cuff compression, and artifacts caused by involuntary movements of the lower limbs [31,32], all of which may prevent the detection of the most ischemic site on the foot. In this study, nine patients (18.0%) could not complete the SPP assessment, whereas SMI was possible in all patients, free of pain or motion artifacts during measurement. The less invasive, more accurate assessment of foot microcirculation by SMI favors its use over SPP.

SMI can also be used to evaluate microcirculation in the floor of an ulcer, as demonstrated in Figure 5. Multiple studies have shown a correlation between microcirculation, as determined by SPP, and wound healing [33,34,35]. However, SPP measurements are limited to the periphery of an ulcer; evaluation of the microcirculation inside the ulcer is not possible. Because SMI is painless, it can be used to observe microcirculation in wound tissue directly; this enables not only the monitoring of the response to treatment but also the prediction of wound healing.

Recently, near-infrared spectroscopy (NIRS) has been attracting attention as a minimally invasive method for evaluating lower-extremity circulation. This technique utilizes the tissue permeability of near-infrared light to measure changes in oxyhemoglobin and deoxyhemoglobin in tissue over time and calculate regional tissue oxygen saturation (rSO_2_) [36]. A finger-worn tissue oximeter that uses NIRS technology has been developed (Toccare: Astem Co., Ltd., Kawasaki, Japan) to measure rSO_2_ values to depths of 0–5 mm below the skin surface. According to several research reports, NIRS enables a painless and simple measurement of rSO_2_ in any area of the foot and reflects microcirculation [37,38,39]. While these advantages are similar to those of SMI, NIRS is difficult to quantify; SMI is superior to NIRS for the quantitative assessment of foot microcirculation.

This study had some limitations. First, the study population was small, and the single-center design might have led to bias with respect to the patients’ backgrounds. Second, since most patients in the CLI group had rest pain and only 13% had wounds, we could not investigate the association between the wound healing process and the increase in SMI-VI. Third, all CLI patients in this study had lesions in the SFA, and 42 out of 54 patients (78%) also had below-the-knee arterial stenoses. The SMI-VI should also be evaluated in CLI attributed to the isolated below-the-knee arterial stenosis. Fourth, in patients with complications related to local infection, acute inflammation may lead to capillary hyperemia-related overestimation using SMI during a microvascular assessment. Fifth, an SMI-VI cut-off value to diagnose an ischemic limb could not be validated. Further studies are needed to determine the SMI-VI cut-off value for the diagnosis of an ischemic limb and its use as an indicator of wound healing.

## 5. Conclusions

SMI enables a quantitative evaluation of foot microcirculation in a simple, rapid, and painless manner. Therefore, SMI can be useful in screening for peripheral arterial disease in the outpatient setting and in determining treatment endpoints during EVT. SMI also has the potential to predict wound healing by observing microcirculation directly in wound tissue. SMI offers a new imaging modality for use in the medical management of CLI.

## Figures and Tables

**Figure 1 diagnostics-12-02577-f001:**
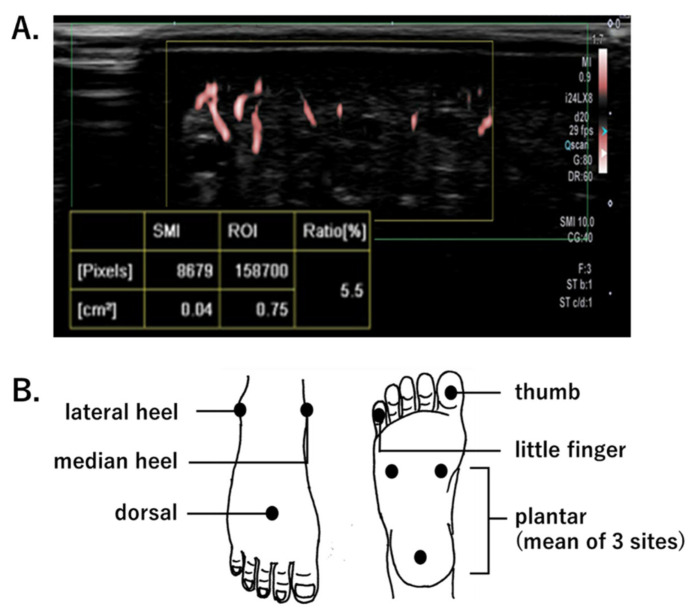
(**A**). Measurement of the superb microvascular imaging-vascular index (SMI SMI-VI), defined as the ratio of color pixels (blood flow signals) to the total pixels in the region of interest (ROI). (**B**). SMI-VI measurement points.

**Figure 2 diagnostics-12-02577-f002:**
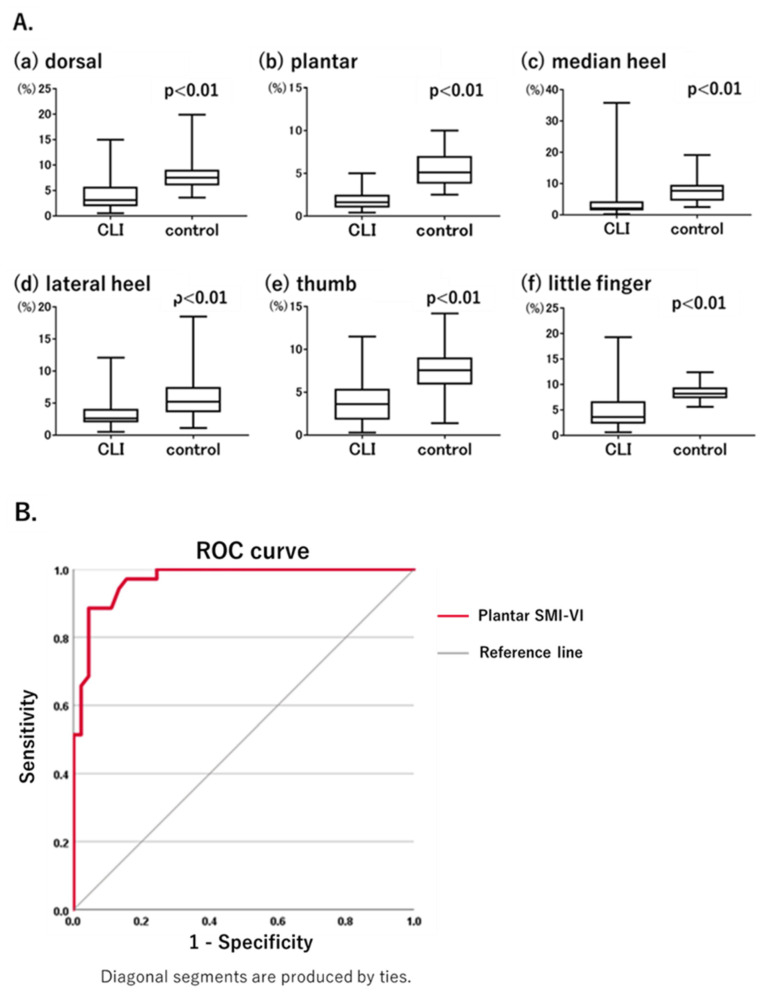
(**A**). Comparison of SMI-VI values between critical limb ischemia (CLI) and control groups. The SMI-VI was significantly lower in the CLI group than in the control group at all measurement points. (**B**). Diagnostic performance of plantar SMI-VI for CLI, as determined using a receiver operating characteristic curve. Area under the receiver operating characteristic curve of 0.969, a sensitivity of 88.6%, and a specificity of 95.6%.

**Figure 3 diagnostics-12-02577-f003:**
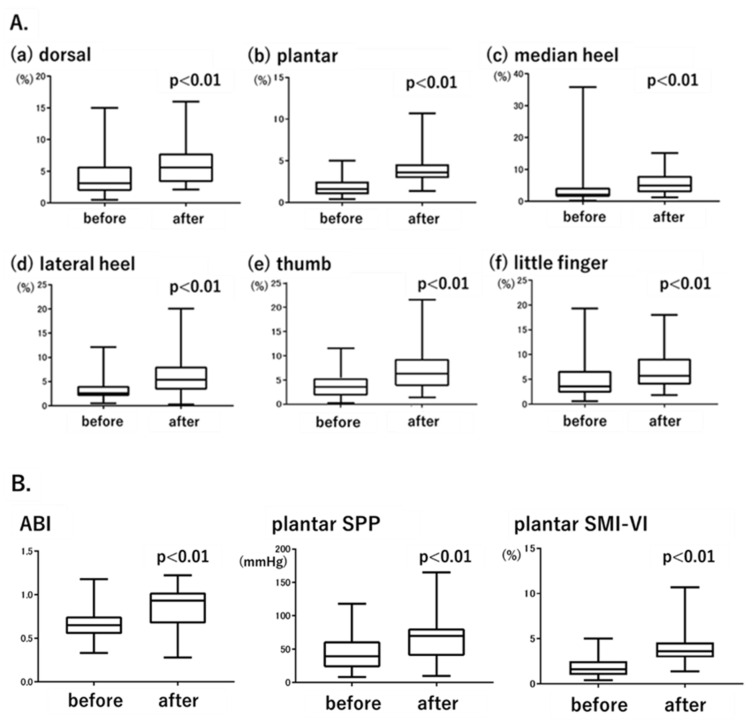
(**A**). Changes in SMI-VI values before and after endovascular treatment (EVT). (**B**). Changes in the ABI, plantar SPP, and plantar SMI-VI before and after EVT.

**Figure 4 diagnostics-12-02577-f004:**
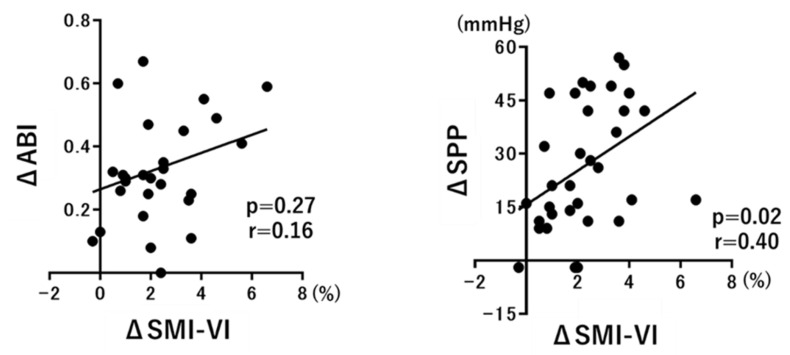
Correlations of ΔSMI-VI with ΔABI and ΔSPP.

**Figure 5 diagnostics-12-02577-f005:**
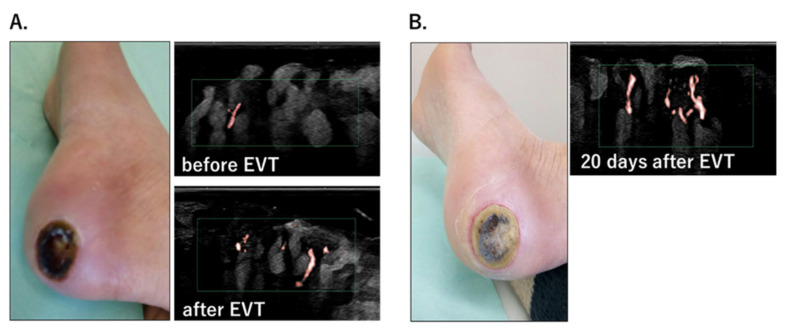
(**A**). Heel ulcer and wound-tissue microcirculation before and after EVT. SMI captured the improvement in microcirculation immediately after EVT. (**B**). The ulcer healed gradually, and abundant microcirculation at the wound bed was maintained 20 days after EVT.

**Table 1 diagnostics-12-02577-t001:** Patient characteristics in the critical limb ischemia (CLI) group.

Patient Characteristics	*n* = 50
Age, yrs	73 (67–79)
Male	36 (72)
Smoking	32 (64)
Ischemic findings	
Rest pain	50 (100)
Ulceration	5 (10)
Necrosis	8 (16)
Wound location	
Toes	4 (8)
Heel	2 (4)
Ankle	1 (2)
Shin	1 (2)
Multiple	5 (10)
Comorbidity	
Hypertension	37 (74)
Dyslipidemia	34 (68)
Diabetes mellitus	30 (60)
Chronic kidney disease	25 (50)
Hemodialysis	11 (22)
Medication	
Aspirin	50 (100)
Cilostazol	26 (52)
Clopidogrel	23 (46)
Calcium channel blockers	28 (56)
ISDN/NTG	7 (14)
PGE_1_/PGI_2_	7 (14)
5-HT_2_ receptor antagonist	8 (16)
Statin	41 (82)

Age is presented as the median (IQR); other values are presented as *n* (%). ISDN, isosorbide dinitrate; NTG, nitroglycerin; PGE1, prostaglandin E1; PGI2, prostaglandin I2; 5-HT2, 5-hydroxytryptamine2.

## Data Availability

The datasets generated during and/or analyzed during the current study are available from the corresponding author on reasonable request.

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
