# Peer review of "Utility of Superb Microvascular Imaging in the Assessment of Foot Perfusion in Patients with Critical Limb Ischemia"

_diagnostics, 2022, doi:10.3390/diagnostics12112577_

Round 1
Reviewer 1 Report
Dear Authors, I have read your manuscript with interest.
The current manuscript titled: "Utility of Superb Microvascular Imaging in the Assessment of Foot Perfusion in Patients with Critical Limb Ischemia" represents an important analysis of evolving field of Cardiology and Vascular Surgery.
The title reflects the manuscript content and helps the reader navigate the article essence.
In my opinion, these are the adjustments which should be made to increase the value of your manuscript:
1. In Introduction chapter, please, add more detail information about peripheral artery disease and its complication – CLI, as well as CLI epidemiology. Also, add please detailed information about superb microvascular imaging use particularities in PAD and CLI.
2. Line 95: change please “m2” to “m2”.
3. Confirm please that the image (Figure 1B) used for this article is original and not plagiarized.
4. Add please Table number in lines 161 and 182.
5. In Results chapter, add please control group characteristics.
6. There is not enough comparative information in the Discussions chapter with other similar studies. Please search for research data and add this information to this section.
7. The Conclusions section provides very brief information, which is the same as the Abstract. Please specify in more detail the need for this diagnostic method and the purpose of its use in everyday practice.
8. The manuscript contains some punctuation errors, please revise the text (e.g., line 213, 277, etc.).
Author Response
We are grateful for the helpful comments and constructive suggestions from the reviewer. We have revised the manuscript accordingly. Our responses to the reviewer’s comments are noted in the word document. Please see the attachment.

Reviewer 2 Report
Actually there is a true need of an efficient and easy tool to evaluate the efficacy of a revascularisation in order to predict acute clinical outcome in terms of wound healing.
Few patients had wounds (the most Rutherford 4, rest pain).
Should be interesting to have a better relationship between SMI-VI improvement and wound healing time and quality.
FROM LINE 113 TO 119: please describe better how could the probe has been positioned in the same point for pre and post intervention comparison.
Author Response

(The authors gave the same response as above.)

Reviewer 3 Report
article is well written and deals with an interesting topic.
references are updated. statistical analysis is reasonable to support conclusions.
manuscript should be considered for publication, in my opinion.
Author Response
Thank you very much for your comments. We arranged professional proofreading by two native speakers of English to correct the English. We attach the certificate.

Round 2
Reviewer 1 Report
I agree with the changes made, which significantly improve the quality of the manuscript. I recommend this article for publication. Good luck!
Author Response
We greatly appreciate the comments and suggestions you gave us.
We will do our best.